# Preparation of a Novel Branched Polyamide 6 (PA6) via Co-Polymerization of ε-Caprolactam and α-Amino-ε-Caprolactam

**DOI:** 10.3390/polym16121719

**Published:** 2024-06-16

**Authors:** Xiaoyu Mao, Wei Liu, Zeyang Li, Shan Mei, Baoning Zong

**Affiliations:** 1Research Center of Renewable Energy, Research Institute of Petroleum Progressing, SINOPEC, Beijing 100083, China; maoxiaoyu.ripp@sinopec.com (X.M.); liuwei2.ripp@sinopec.com (W.L.); lizeyang.ripp@sinopec.com (Z.L.); 2No. 22 Research Department, Research Institute of Petroleum Progressing, SINOPEC, Beijing 100083, China; meishan.ripp@sinopec.com; 3State Key Laboratory of Catalytic Material and Reaction Engineering, Research Institute of Petroleum Progressing, SINOPEC, 18th Xueyuan Road, Haidian District, Beijing 100083, China

**Keywords:** hydrolytic ring-opening polymerization, modified branched PA6, thermal properties, rheological properties, mechanical properties

## Abstract

In this study, a novel branched polyamide 6 has been synthesized via the hydrolytic ring-opening co-polymerization of ε-caprolactam (CPL) and α-Amino-ε-caprolactam (ACL). The NMR characterization proves the existence of a branched chain structure. The rheological test determines that there is a remarkable increase in the melt index (MFR), zero shear rate viscosity, and storage modulus in the low-frequency region. The shear-thinning phenomenon becomes more obvious. The thermal properties tested by differential scanning calorimetry (DSC) show that the melting point and crystallinity of co-polymers decrease with the incorporation of ACL. However, the crystal structure of the samples only exhibits a slight change. When the ACL content in the feed is 1 wt%, the tensile strength and fracture elongation rate of the co-polymers show a significant enhancement.

## 1. Introduction

Polyamide 6 (PA6), also known as nylon 6, is a kind of widely used polymer with processability, excellent mechanical properties, chemical durability, and low cost. It can be used for spinning, blow molding, and industrial molding, which contribute to a wide range of applications in the fields of textiles, packaging, and engineering plastics [1,2,3]. Currently, industrial PA6 is mostly prepared by the hydrolytic ring-opening polymerization of ε-caprolactam (CPL) at high temperature, which has the advantages of relatively mild reaction conditions, easy scale-up production, and narrow molecular weight distribution. However, due to the influence of the hydrolytic ring-opening chemical equilibrium, the molecular weight of industrial PA6 is not high [1,4]. Linear PA6 with low molecular weight has an insufficient melt strength, which limits its application in thermal forming, injection molding, foaming, and other fields dominated by stretching flow [5]. In addition, linear PA6 with low molecular weight is highly sensitive to crack propagation and exhibits brittle fracture behavior under low temperature and severe load conditions, with poor impact resistance [6]. By introducing multifunctional monomers into the polymerization process, branched PA6 can be prepared. Compared with linear PA6, branched PA6 has significantly different melt rheological properties and crystallization properties, which affects its processing properties in spinning, film stretching, injection molding, and other processes, thereby expanding the applications of PA6 [7,8].

Branched PA can be divided into star branched PA and randomly branched PA, according to the different branching sites [9]. Star branched PA typically has a multifunctional branched core. The monomers polymerize into chains on the functional groups of the branched core. Star branched PA has unique properties such as low crystallinity, low melt viscosity, high molecular surface functionality, and small fluid dynamic volume. Randomly branched PA can be regarded as a multi-level star branched PA, and the generation of its branches can significantly increase the zero shear viscosity and strength of polymer melts and strengthen the tensile strain hardening behavior, which is particularly beneficial for the application of materials in flow fields dominated by tension, such as blowing and spinning [10]. Randomly branched PA6 can be obtained by the co-polymerization of CPL and multifunctional co-monomers, such as multifunctional amine or carboxylic acid. However, the degree of polymerization in the polycondensation reaction is relatively low, and the branch content of the prepared randomly branched PA6 is correspondingly not high enough. Scholl et al. [11] prepared highly branched PA6 with a relative molecular weight of 23,000 by starting from ε-lysine salt and adding a certain amount of co-polymerizable monomer and adjusting the polymerization conditions. Steeman et al. [12] prepared randomly branched nylon 6 by co-polymerizing 2,4,6-triaminohexanoic acid with CPL, and the long-chain branching led to an increase in the zero-shear viscosity of PA6 and a more pronounced shear-thinning behavior, while the melt strength increased. Li et al. [13] obtained a randomly branched PA6 with a small amount of long-chain via co-polymerization of ε-lysine and CPL. With the close relative viscosity, the melting point of long-chain branched PA6 is closed to that of linear PA6, but it exhibits a more significant shear-thinning phenomenon. The introduction of long-chain branching increases the entanglement between molecular chains, making the molecular chains more easily oriented along the processing direction. However, due to the low content of branched chains, the changes in the mechanical properties of the co-polymer are not significant.

By co-polymerizing ε-lysine or its derivatives with CPL, a novel branched PA6 can be obtained. In our previous work, we successfully synthesized α-Amino-ε-caprolactam (ACL) by cyclizing lysine [14,15]. ACL has a similar structure to CPL, but an amino group at the α position contributes to bifunctionality after hydrolysis. In this study, we co-polymerized CPL and ACL through hydrolytic ring-opening polymerization under different ACL contents and successfully prepared a modified branched PA6, named P(ACL-*co*-CPL). The mechanism of this polymerization process is similar to that of the homopolymerization of CPL, and the process of branching is an addition reaction of monomers rather than a polycondensation reaction. Therefore, high-branching-content PA6 can be obtained. The structure of P(ACL-*co*-CPL) was characterized by ^1^H NMR and ^13^C NMR spectra, which confirmed the feasibility of co-polymerization and the generation of branched chains. The rheological properties of P(ACL-*co*-CPL) were tested by using the rotational rheometer, and its thermodynamic characteristics were determined by using the differential scanning calorimetry (DSC). Finally, the tensile behavior of P(ACL-*co*-CPL) was tested, and the optimal ratio of caprolactam to ACL was determined based on the results of rheological and thermodynamic tests.

## 2. Materials and Methods

### 2.1. Materials

The CPL used in this study was supplied by Baling Petrochemical Company, SINOPEC (Yueyang, Hunan, China), and the ACL was synthesized in our earlier work (Beijing, China). Sulfuric acid and 2,2,2-Trifluoroethanol was purchased from Acros (Beijing, China). The hydrochloric acid ethanol standard solution and potassium hydroxide ethanol standard solution used in this study were provided by Alfa (Shanghai, China).

### 2.2. Co-Polymerization of CPL and ACL [13]

A certain ratio of CPL, ACL, and water were added to a 5 L autoclave (Keli Chemical Equipment Co., Ltd., Yantai, China). Vacuuming was applied at room temperature until the pressure in autoclave reached −0.05 MPa, then N_2_ gas was inflated to relieve the vacuum. This vacuuming–inflation cycle was repeated 3 times, and finally we used inflating N_2_ gas to bring the pressure in autoclave to around 0.05 MPa. The temperature was initially set at 225 °C, and stirring was started at 100 °C. After pre-polymerizing at 225 °C for 0.5 h, the temperature was raised to 245 °C and maintained for 3 h under pressure. The pressure was then released to atmospheric pressure, and the reaction was continued for another 1 h. Finally, vacuuming was applied until the pressure reached below −0.05 MPa, and the reaction was continued for 1 h. The product was then discharged and cut into pellets. The pelleted co-polymer was dissolved in 90 °C hot water in order to remove the residual monomers and low molecular weight polymers.

The reaction mechanism of this system is similar to that of preparation of PA6 via hydrolytic ring-opening polymerization of caprolactam. Under conditions of high temperature and water presence, the amide bond of CPL undergoes cleavage, resulting in the formation of aminocaproic acid (Figure 1a). Aminocaproic acid can undergo condensation to form a short-chain polyamide. The terminal amino group of the short-chain polyamide attacks the carbonyl carbon of the CPL monomer’s amide bond, directly adding the CPL monomer to the polymer chain and achieving chain growth [16]. For our co-polymer system, the reaction progress and mechanism are shown in Figure 2. ACL was supposed to have similar properties to CPL so that its amide bond can also undergo cleavage to form a linear molecule (Figure 1b). The polycondensation of ring-opening products led to the formation of low molecular weight polymers with a main chain and side chain (Figure 1c). The terminal amino groups in both the main chain and side chain of the polymer can react with the monomers protonated by carboxyl groups leading to chain addition (Figure 1d).

### 2.3. Injection Molding of Co-Polymer

The prepared co-polymers were dried in a vacuum oven at 60 °C for 12 h to remove the absorbed moisture in the air. The barrel temperature of the injection molding machine (MiniJet Pro, Thermo Fisher Sci., Waltham, MA, USA) was set to 240 °C, and the temperature of the mold cavity was set to 100 °C. The injection time was 15 s and the injection pressure was 800 bar. The holding time was 10 s, and the holding pressure was 650 bar.

### 2.4. Characterization

The relative viscosity (*η*_r_), intrinsic viscosity ([η]), and viscosity-average molecular weight of the co-polymer were determined by using a Ubbelohde viscometer (IV2000, Zhuoxiang Technology Co., Ltd., Hangzhou, China). The co-polymer was dissolved in a 96 wt% sulfuric acid solution (solute:solvent, 1 g:100 mL). The time it took for the solvent and co-polymer solution to pass through the capillary at 25 °C was measured using the Ubbelohde viscometer. The molecular weight of the co-polymer was calculated using the Mark–Houwink equation [η] = k[M_w_]^a^. The Mark–Houwink constants for nylon 6 in a 25 °C, 96% sulfuric acid solution are k = 6.3 × 10^−4^ and a = 0.764 [1].

The end-group content of the co-polymer was tested by using a potentiometric titrator (916T-touch, Metrohm, Herisau, Switzerland) according to GB/T 38138-2019 [17]. The co-polymer was dissolved in an 88% trifluoroethanol solution at 60 °C. First, 0.02 mol/L hydrochloric acid–ethanol standard solution was used to titrate the polymer solution cooled to 25 °C, and a blank test was performed. Then, a 0.02 mol/L potassium hydroxide–ethanol standard solution was used to titrate the solution that had just undergone amino concentration titration, with the titration process first neutralizing the excess hydrochloric acid and then titrating to the endpoint.

The structures of the resultant nylon-6 polymers were investigated by ^1^H NMR and ^13^C NMR (AVANCE NEO 500 M, Bruker, Billerica, MA, USA), using formic acid/trifluoroacetic acid-d (1:1, *v*/*v*) as the solvent.

The rheological properties of the samples were measured by using a rotational rheometer (P25CSL, HAAKE MARS, Waltham, MA, USA). The test specimens were prepared by injection molding with a diameter of 20 mm and a thickness of 1 mm. A frequency sweep was performed under oscillatory mode with a strain of 1% and a frequency range of 0.1–500 rad/s to obtain the variation in the complex viscosity (η), storage modulus (G′) and loss modulus (G″) with angular frequency (ω). The melt flow index of the co-polymers was determined using a melt flow indexer (MI40, GOETTFRT, Essen, Germany) at a testing pressure of 2.16 kg and a chamber temperature of 230 °C.

The thermal properties of the co-polymers were determined by differential scanning calorimetry (DSC3, METLER TOLEDO, Zurich, Switzerland). Under a N_2_ atmosphere, the samples were first heated from 25 °C to 300 °C to remove thermal history, with a heating rate of 20 °C/min. Then, at a cooling rate of 10 °C/min, the samples were cooled from 300 °C to 25 °C to test their crystallization performance. Finally, the samples were heated from 25 °C to 300 °C at a heating rate of 10 °C/min to test their melting performance. The degree of crystallinity (Xc) was obtained by the following equation.
*X*_c_ = Δ*H*_m_/Δ*H*_0_ × 100%(1)
where Δ*H*_m_ is the specific enthalpy of melting, and Δ*H*_0_ is the enthalpy of melting with 100% crystalline nylon 6 (188 J/g) [18]. 

The crystal type of the sample was tested using an X-ray diffractometer (Empyrean, PANALYTICAL, Alemlo, The Netherlands), with a scanning range of 5° to 50° and a scanning speed of 2°/min.

The morphology of the cross-section of co-polymers was observed by electron microscopy (S-4800, Hitachi, Tokyo, Japan) with a test voltage of 5 V.

The tensile mechanical properties were checked by a universal tensile machine (CMT2000, MTS/SANS, Eden Prairie, MN, USA) according to ASTM D638 [19]. The strain rate was 10 mm/min.

All of these specimens were kept in a desiccator under vacuum for 24 h before the measurements.

## 3. Results and Discussion

The chemical shifts of H atoms and C at branching sites were determined by ^1^H NMR and ^13^C NMR, respectively, to determine the structure of the P(ACL-*co*-CPL) co-polymer. The structural formula and ^1^H NMR spectrum of P(ACL/CPL) are shown in Figure 2a, with the chemical shifts of H atoms as follows: δ4.63 (branching site), δ3.42, 2.58, 1.45–1.78 (main chain and side chain). The structural formula and ^13^C NMR spectrum of P(ACL-*co*-CPL) are given in Figure 2b, with the chemical shifts of C atoms as follows: δ65.2 (branching site), δ3.42, 2.58, 1.45–1.78 (main chain and side chain).

The end-group test results of the co-polymers are shown in Table 1. When the ACL content in the feed was less than 1 wt%, the carboxyl group concentration was similar, around 65 mmol/kg. When the ACL content exceeded 1 wt%, the carboxyl group concentration increased, which was due to a significant decrease in molecular weight. Meanwhile, the end-amine concentration as well as the ratio of end-amine to end-carboxyl groups, significantly increased with the incorporation of ACL. Based on the mechanism of branching mentioned above, this is because the incorporation of ACL increased the number of branches, and each branch end is also an amino group, resulting in a significant increase in the amino group concentration.

The rheological properties of polymers, highly sensitive to changes in the molecular chain, can be used to reveal the chain structure of polymers [20]. The complex viscosity (η) of pure PA6 and our co-polymer samples as a function of angular frequency (ω) is shown in Figure 3. With a linear molecular chain structure, pure PA6’s complex viscosity showed little change as the angular frequency increased, and the η-ω curve showed a clear Newtonian plateau. After the incorporation of ACL, the complex viscosity of the co-polymers decreased significantly with the increase in angular frequency and the Newtonian plateau disappeared, as the shear-thinning phenomenon occurred. Zero shear viscosity is the complex viscosity of a sample when the shear rate tends to zero and the system approaches the equilibrium state. It can reflect the structural characteristics of the sample in the equilibrium or near-equilibrium state [21]. For linear polymers, the zero shear viscosity is related to the relative molecular weight of the sample, and the relationship between the relative viscosity of PA6 and the relative molecular weight follows a power law of 3.4 [22] (shown in Equation (2)).
η_0_ = *k*M_w_^3.4^(2)
where η_0_ represents the zero shear viscosity, M_w_ represents the relative molecular weight, and *k* represents a temperature-dependent constant. From the law, we can see that a larger zero shear viscosity corresponds to a higher molecular weight. However, with the incorporation of ACL, the molecular weight of the co-polymers first remained unchanged and then decreased. This indicates that the structure of the co-polymers is no longer linear. The zero shear viscosity of the co-polymers can be calculated through the simple Carreau equation with the Cox–Merz rule [23]
η(γ)/η_0_ = (1 + (γτ_n_)^2^)^(n−1)/2^(3)
where η_0_ is the zero shear rate viscosity, γ is the shear rate, τ_n_ is the characteristic time, and n is a parameter. As the ACL content in the feed exceeded 1 wt%, the zero shear viscosity of the co-polymer increased significantly. This is because under near-equilibrium conditions, the viscosity of the material is mainly influenced by long branches with longer relaxation times. The entanglement of long branches hinders the movement of chain segments, resulting in an increase in the zero shear viscosity of long-branched samples compared to linear chains with the same molecular weight [24,25,26,27]. The generation of a gel structure can be another possible reason for the significant increase in zero shear viscosity. Although three-dimensionally crosslinked polymers are incapable of macroscopic viscous flow, the crosslinked chains can flow with other chains when the ACL content is very small. However, it was checked that all of the co-polymer is completely soluble in formic acid with no remaining particles by the particle size analyzer. In addition, the crosslinked section is infusible in the melt, but the melt we observed in the experiment was all homogeneous [28].

The storage modulus, loss modulus, and loss factor are also important rheological parameters reflecting the structure of the sample. The higher the storage modulus, the stronger the ability of the sample to maintain its original equilibrium state after being subjected to an external force, and the longer the time it takes to return to its original equilibrium shape after the same deformation, which can also be referred to as the relaxation time. On the other hand, the loss modulus reflects the viscous response of the sample, with a higher loss modulus indicating stronger liquid-like behavior [29]. According to linear viscoelastic theory, in the low-frequency region, pure linear PA6 exhibited terminal behavior; that is, the storage modulus and loss modulus have the following relationship with angular frequency [30]:G′ ∝ *ω^2^*, logG″ ∝ 2log*ω*
G″ ∝ *ω*, logG″ ∝ log*ω*

According to the relationship, in the double logarithmic coordinate system, the slope of the linear pure PA6 G′-ω curve approaches 2 in the low-frequency region. As shown in Figure 4a, the slope of the PA6 curve was consistent with the above, while with the addition of cycloaliphatic lysine, the slope of the G′-ω curve in the low-frequency region for our co-polymers gradually decreased from 1.75 (0.1 wt% ACL content in feed) to 0.17 (5 wt% ACL content in feed), deviating from the linear end behavior. Meanwhile, the storage modulus value increased by nearly 200 times, reflecting that with the addition of ACL, the long-chain branching structure in the co-polymers increased, the degree of molecular entanglement raised, and the sample’s solid-like properties were enhanced, corresponding to a longer relaxation time.

The loss factor tanδ is the ratio of the loss modulus to the storage modulus (tanδ = G″/G′), and its inverse trigonometric function is the loss angle. The formation of branched chains is also reflected in the changes in the loss factor and loss angle [31]. Figure 5 shows the tan δ-ω curves for pure PA6 and our co-polymers with different contents of ACL. For pure PA6, the loss factor tanδ increased with the decrease in ω, and the curve raised sharply at the low-frequency end, which was a typical liquid-like terminal behavior of linear molecular chains. With the addition of ACL, the tanδ decreased and a plateau formed in the low-frequency region. This is consistent with the findings of Graebling et al. [32], who believed that the storage modulus of the sample is more sensitive to the formation of branches, and a small amount of branching can cause a significant increase in the storage modulus, while the loss modulus is less sensitive to this, hence the magnitude of tanδ is mainly influenced by changes in the storage modulus. With the formation of branched chains, there is a significant increase in the storage modulus in the low-frequency region, while the change in the loss modulus is relatively small; therefore, tanδ begins to decline in the low-frequency region. At the same time, the presence of branches enhances the relaxation time of the sample at the low-frequency end, and the storage modulus of the sample gradually stabilizes, hence the tan δ-ω curve shows a plateau.

The Cole–Cole plot represents the relationship between the imaginary viscosity η″ (η″ = G′/ω) and the real viscosity η′ (η′ = G″/ω). The deflection of the Cole-Cole plot curve can be used to characterize the relaxation time and relaxation mechanism of polymer samples [33]. For linear chain structures, the Cole-Cole curves are close to semicircular and with the generation of branched chains, the Cole-Cole curve of the co-polymers gradually deviates from the semicircular shape and begins to turn up in the low-frequency region. This indicates that the relaxation process of the sample is extended and the relaxation time of the system increases. Figure 5b exhibits the Cole–Cole plot of our co-polymers, which is consistent with the above content. 

The thermal properties of the co-polymers were determined by DSC scanning, as shown in Figure 6a,b and Table 2. When the ACL content in the feed was below 1 wt%, the melt peak was sharp and exhibited a clear peak near 220 °C. However, a significant decrease in the melting point occurred when the ACL content exceeded 1 wt%. The crystallization process of the samples was more sensitive to the addition of ACL, as the crystallization temperature and degree of crystallinity decreased significantly and the crystallization peak became broader. This is because the addition of a small amount of ACL disrupts the regularity of the molecular chains, and the presence of some shorter side chains reduces the chain migration rate, hindering the crystallization of the samples [34]. When the ACL content in the feed was 5 wt%, the co-polymer exhibited a double crystallization peak, which may be due to the different crystallization rates of different segments in the co-polymer. As the incorporated ACL increased, the regularity of the co-polymer chains’ structure became further disrupted, as well as the content of short branched chains increasing. This contributed to a decrease in the crystallization rate of some chains in the co-polymer. The crystallization of polymers is a process in which molecular chains are arranged from disordered to ordered [35]. At a higher cooling rate, the residence time of the polymer at each temperature correspondingly becomes shorter. Some chains which are difficult to crystallize do not have time to make changes, and the folding and arrangement of the chains are incomplete, leading to crystallization hysteresis. Therefore, we conducted crystallization tests on the co-polymer at different cooling rates, and the cooling curves obtained are shown in Figure 6c. When the cooling rate increased to 20 °C/min, the double crystallization peak became more pronounced, while when the cooling rate decreased to 5 °C/min, the crystallization peak became a single peak. This indicates that the double peaks observed in the cooling crystallization curve at higher cooling rates are due to the difficulty of chains in responding to faster cooling rates, resulting in crystallization hysteresis. Figure 6d shows the X-ray diffraction characterization results of the co-polymers with different ACL contents. It can be seen that although the incorporation of ACL affects the crystallinity of the samples, the crystal structure of the samples showed limit change, and all samples exhibit characteristic peaks at 20.4° and 23.9°, corresponding to the α-crystal form of PA6.

The SEM results of pure PA6 and P(ACL-*co*-CPL) cross-sections are shown in Figure 7. It can be seen that the cross-sectional morphology of the co-polymers did not change significantly after the incorporation of ACL. It is speculated that the ACL and PA6 chains have very similar structures, leading to a good compatibility. In addition, the crystal form of the co-polymers remained unchanged after the incorporation of ACL, which is also the main reason for the similar cross-sectional morphology.

Figure 8 shows the stress-strain curve during the tensile process of co-polymers with different ACL contents. It can be seen that the stress decreased after the yield point of pure nylon samples, and then remained constant until the sample fractured, which was the tensile strength. With the addition of ACL, the stress plateau after yield became shorter, and then increased with strain, experiencing a 2–4 times increment. The tensile mechanical properties of PA6 and P(ACL-*co*-CPL) can be seen in Table 3. When the ACL content did not exceed 1 wt%, the tensile strength of the samples increased with the ACL content, and it was always greater than the yield strength, indicating the occurrence of strain hardening. When the ACL content was 2%, the stress after yield also increased multiple times with strain, but the ultimate fracture strength was lower than the yield strength, indicating no strain hardening phenomenon. When the ACL content was 5%, the sample fractured quickly after yield. With the addition of ACL, the change in the co-polymers’ yield strength was not significant, but the tensile strength and fracture elongation rate significantly increased when the ACL content was below 1 wt%, indicating an enhanced toughness of the samples. After the ACL content exceeded 1 wt%, the tensile strength and fracture elongation rate decreased. Typically, for linear polymers, the tensile strength and fracture elongation rate decreased with decreasing crystallinity. The addition of ACL reduced the crystallinity but increased the branching, resulting in more chain entanglements. During the tensile process, the entangled chains rearranged and gradually untangled after yield, requiring further energy barrier overcoming, thus the stress gradually increased [36]. When the ACL content exceeded 2%, the molecular weight of the co-polymers started to decrease significantly, leading to a decrease in the subsequent tensile strength and fracture elongation rate. Comparing the tensile properties of co-polymers with different ACL contents, P(ACL/CPL, 1%) exhibited the best tensile mechanical properties.

## 4. Conclusions

In this study, we successfully synthesized a novel branched PA6 with ε-caprolactam (CPL) and α-Amino-ε-caprolactam (ACL) via hydrolytic ring-opening co-polymerization. The polymerization mechanism is similar to the homo-polymerization of CPL. The met index (MFR), zero shear rate viscosity, and storage modulus in the low-frequency region showed a remarkable increase. The shear-thinning phenomenon became more obvious. The melting point and crystallinity of co-polymers decreased with the increase in ACL incorporation. However, the crystal structure of the samples only showed a slight change. The incorporation of ACL can improve the tensile mechanical properties of the co-polymers. When the ACL content in the feed is 1%, the tensile strength and fracture elongation rate of the co-polymers show the most significant increase. Our research contributes to the development of more effective polymer materials.

## Figures and Tables

**Figure 1 polymers-16-01719-f001:**
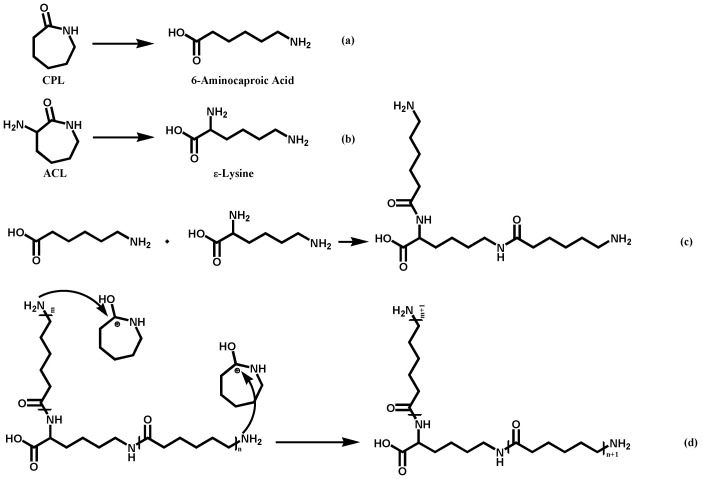
The co-polymerization progress of CPL and ACL: (**a**) The open-ring of CPL; (**b**) The open-ring of ACL; (**c**) The poly-condensation of open-ring products; (**d**) The chain addition.

**Figure 2 polymers-16-01719-f002:**
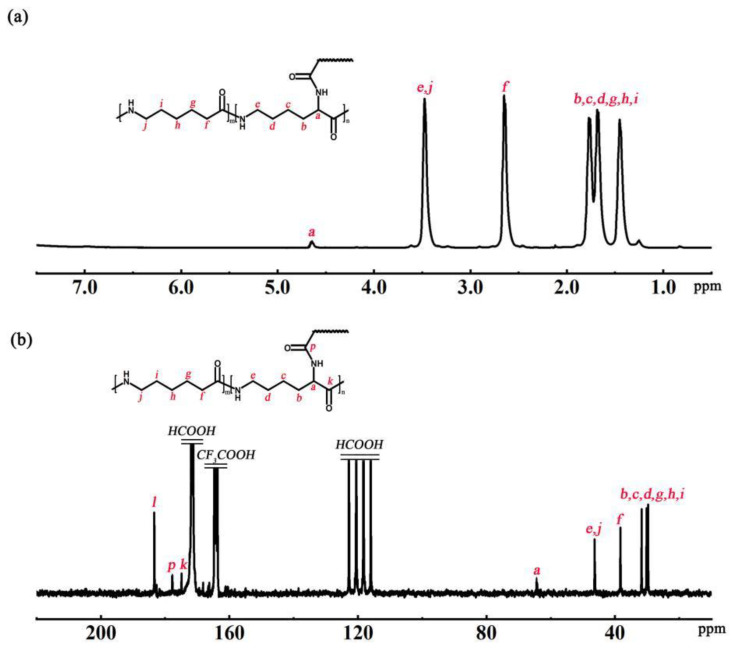
The ^1^H NMR spectra (**a**) and the ^13^C NMR spectra (**b**) of P(ACL-*co*-CPL).

**Figure 3 polymers-16-01719-f003:**
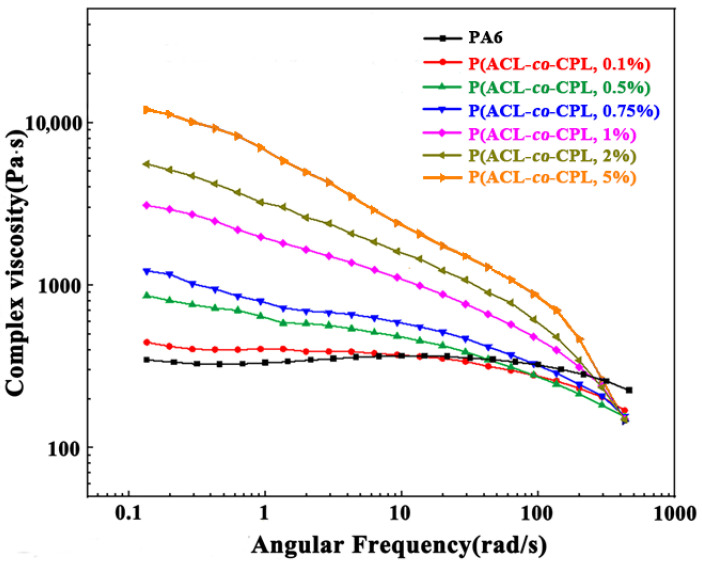
Complex viscosity of PA6 and P(ACL-*co*-CPL).

**Figure 4 polymers-16-01719-f004:**
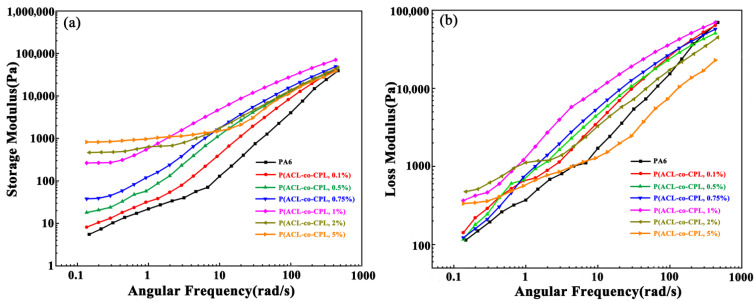
Storage modulus (**a**) and loss modulus (**b**) of PA6 and P(ACL-*co*-CPL).

**Figure 5 polymers-16-01719-f005:**
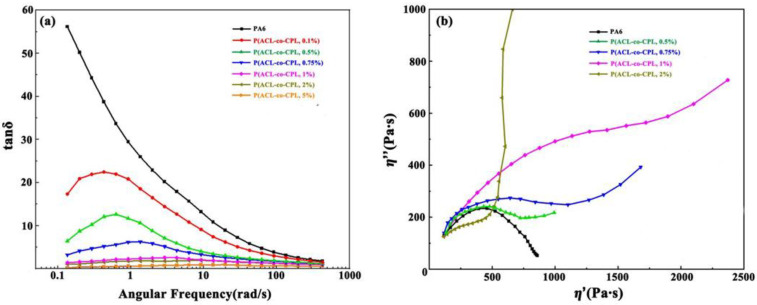
(**a**) Loss factor of PA6 and P(ACL-*co*-CPL); (**b**) Cole–Cole plot of PA6 and P(ACL-*co*-CPL).

**Figure 6 polymers-16-01719-f006:**
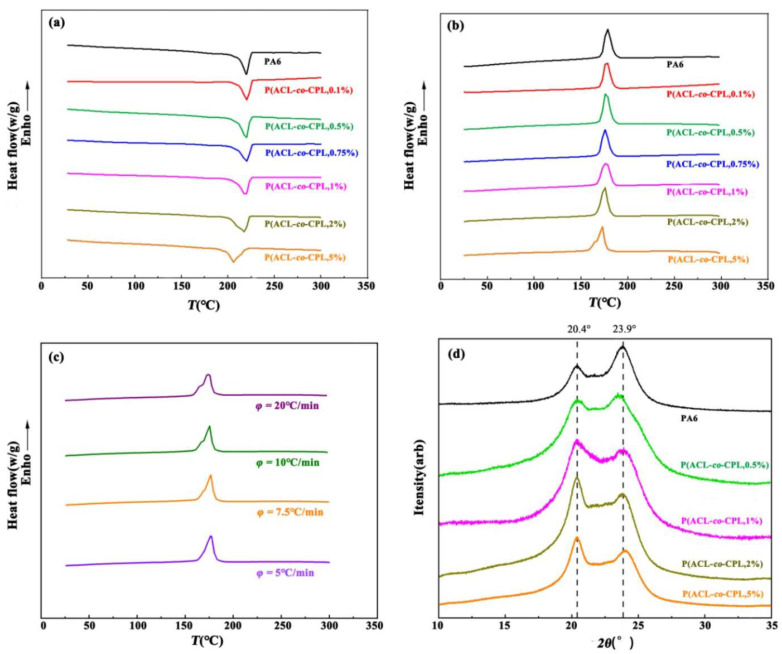
(**a**) Melting curves of PA6 and P(ACL-*co*-CPL); (**b**) Crystallization curves of PA6 and P(ACL-*co*-CPL); (**c**) Crystallization of P(ACL-*co*-CPL) with 5 wt% ACL content in different cooling rate; (**d**) XRD spectra of PA6 and P(ACL-*co*-CPL).

**Figure 7 polymers-16-01719-f007:**
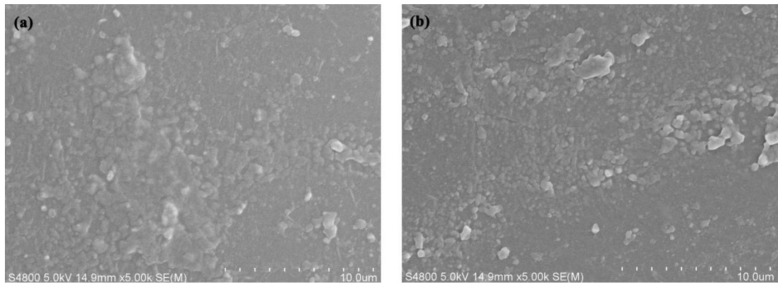
Cross-sectional morphology of PA6 (**a**) and P(ACL-*co*-CPL) (**b**).

**Figure 8 polymers-16-01719-f008:**
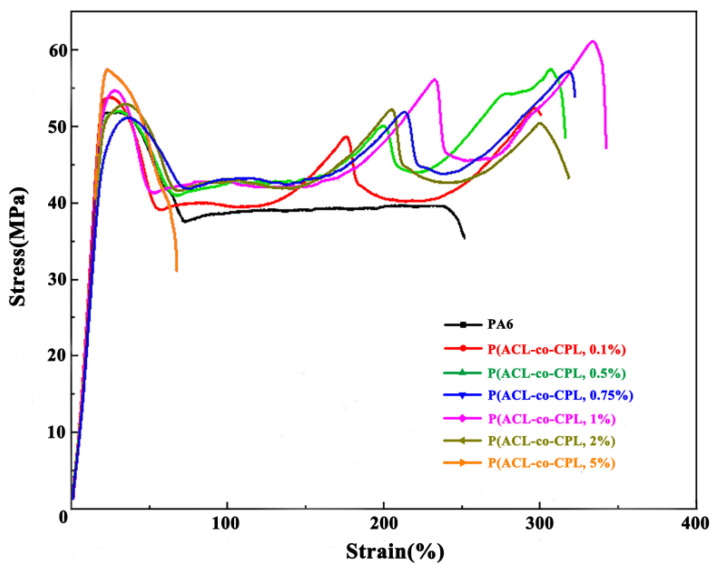
Tensile curve of PA6 and P(ACL-*co*-CPL) co-polymers.

**Table 1 polymers-16-01719-t001:** The basic properties of PA6 and P(ACL-*co*-CPL).

Entry	Feed ACL Content Percentage Mass Ratio	Relative Viscosity	Molecular Mass	End Group Concentration Amino/Carboxyl (mmol/kg)	Melt Index	Zero Shear Viscosity (Pas)
1	0	2.524	16,590	55.06/64.51	23.74	324
2	0.1%	2.510	15,850	59.06/65.98	32.58	425
3	0.5%	2.486	15,640	71.94/65.11	45.68	1075
4	0.75%	2.454	15,220	83.48/66.26	44.32	1098
5	1%	2.424	14,910	96.95/65.66	50.68	3028
6	2%	2.198	12,280	153.4/77.26	92.42	7256
7	5%	1.897	9240	285.4/92.24	150.2	10,680

**Table 2 polymers-16-01719-t002:** The thermal properties of PA6 and P(ACL-*co*-CPL).

Entry	Feed ACL Content Percentage Mass Ratio	Melting Point (°C)	Crystallization Temperature (°C)	Crystallinity (%)
1	0	221	184	27.38
2	0.10%	220.6	177.1	25.59
3	0.50%	219.7	176	25.56
4	0.75%	219	175.3	25.32
5	1%	218.8	175.7	25.19
6	2%	216.7	174.9	24.2
7	5%	206.2	172.3	22.56

**Table 3 polymers-16-01719-t003:** The tensile mechanical properties of PA6 and P(ACL-*co*-CPL).

Entry	Feed ACL Content Percentage Mass Ratio	Yield Strength (MPa)	Tensile Strength (MPa)	Fracture Elongation Rate (%)
1	0	51.55	42.52	250
2	0.10%	51.93	50.12	300
3	0.50%	52.07	55.42	312
4	0.75%	52.66	58.84	318
5	1%	52.87	62.27	358
6	2%	54.25	51.92	321
7	5%	58.2	58.2	66.7

## Data Availability

Data are contained within the article.

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
