# Peer review of "Preparation of a Novel Branched Polyamide 6 (PA6) via Co-Polymerization of ε-Caprolactam and α-Amino-ε-Caprolactam"

_polymers, 2024, doi:10.3390/polym16121719_

Round 1
Reviewer 1 Report
Comments and Suggestions for Authors
This study has been focused on preparing a novel branched polyamide 6 via copolymerization of ε-caprolactam (CPL) and α-amino-ε-caprolactam (ACL) through hydrolytic ring-open co-polymerization. The NMR spectral analysis and rheological test proved formation of branched structures. The formation of these branched copolymer chains onto polyamide 6 caused a clear change in its rheological properties. The melt flow index (MFR), zero shear rate viscosity and storage modulus at low frequency region increased remarkably. The shear thinning phenomenon became more noticeable. Characterization of thermal properties by differential scanning calorimetry (DSC) showed that the melting point and crystallinity percentage of the resulting co-polymers decreased with increasing the co-monomeric ACL proportion but, its crystal form didn’t change. Utilization of the proper amount of ACL led to improve the mechanical properties of the obtained co-polymers. Where the highest increase of the tensile strength and elongation percent of the co-polymers exhibited at using ACL at percentage of 1%. This study could be reconsidered for publication in Polymers after following major revision;
o There are many linguistic corrections should be done, particularly in punctuations and rephrasing sentences to elucidate the meaning. Also, there are some grammatical mistakes should be corrected. Ensure that all your statements are justified and the strength of language is appropriate. In scientific writing, you should use passive voice to focus on the action or result.
o Relative lack of novelty.
o Title of this research article does not accurately reflect the content; therefore, I suggest entitle it as follow;
“Development of A Novel Branched Nylon 6 via Ring-Opening Polymerization of ε-Caprolactam and α-Amino-ε-caprolactam”
“Hydrolytic Ring-Opening Polymerization of ε-Caprolactam and α-Amino-ε-caprolactam for Developing a Novel Branched Polyamide 6”
“Preparation of A Novel Branched Polyamide 6 (PA6) via Copolymerization of ε-Caprolactam and α-Amino-ε-caprolactam”
o Abstract section should be re-organized and improved.
o The introduction section should be enriched with a lot of knowledge that serve and elucidate the importance of this study and mentioning the other previous studies addressing synthesis and properties of polyamide 6 based on different lactams illustrating the unique of this polymer.
o The synthesized copolymer should be called poly (ε-caprolactam-co-α-amino-ε-caprolactam P(ACL-co-CPL).
o In the copolymerization of ε-caprolactam and α-amino-ε-caprolactam section;
§ Mention the reference of synthesis methods
§ Relocate the copolymerization mechanism from the results to this section.
§ The copolymerization procedures and conditions play a major role in determining the structural properties of the resulting copolymers and these different architectures of copolymers may attribute interesting properties. Therefore, unification to these procedures and conditions in each synthesis time based on batch technique should be assured. Indicate how?
§ The authors have mentioned that resulting copolymers were dissolved in 90℃ hot water in order to remove the residual monomers and low molecular weight polymers. This is an immersion not a dissolution.
o The1H and 13C NMR spectral analyses should be taken place for all the resulting copolymers prepared using different co-monomeric ratios. Moreover, the multivariant analysis should be applied to the obtained 13C NMR spectra to evaluate the chemical compositions and degree of branching.
o The authors use one variable through this study which was wrongly called feed CPL/ACL mass ratio. The presented values are not co-monomeric ratios, they are proportions of ACL (as a percent of CPL).
o The copolymers specimens should be characterized by SEM and AFM techniques.
o In “Results” section, first of all this is “Results and Discussion”. It should be improved
o There is an elongation and repeating in the characterization of rheological properties on the expense of the characterization of other physical properties (thermal and mechanical)
o Conclusion section must be improved and shorten. In the final section “Conclusion”, the authors should be emphasized the key findings of the work and its general significance, indicating clearly how this study has developed branched PA 6. i.e., the final decision about your work firmly based on the data presented. Moreover, conclusion could include the recommendation and proposed future work.

Reviewer 2 Report
Comments and Suggestions for Authors
Dear Authors,
I have completed the review of your manuscript titled "Preparation and properties of a novel Nylon 6 co-polymerized by ε-caprolactam and α-Amino-ε-caprolactam". However, I would like to recommend some revisions before further consideration.
1. It would be helpful to provide more details about the hydrolytic ring-open co-polymerization. Were any catalysts or reaction conditions employed?
2. What are the underlying principles that govern the shape of the Cole-Cole plot? In Figure 7, you mentioned that the Cole-Cole curve of PA6 is close to semicircular. Can you explain why a semicircular curve is expected for linear molecular chain structures?
3. Please provide more insight into how the presence of shorter side chains affects the chain migration rate and overall crystallization behavior.
4. For Figure 8(a)(b) and Table 2, Could you elaborate on the implications of the observed changes in the thermal properties of the co-polymers (including the melting point, crystallization temperature, and degree of crystallinity) when ACL is added? How do these changes relate to the disruption of molecular chain regularity and the presence of branching?
5. You mentioned a double crystallization peak observed in the co-polymer with 5% ACL content. What are the possible reasons for this phenomenon? How does the cooling rate affect the crystallization behavior, and why does the double peak become more pronounced at higher cooling rates?
6. It would be helpful to explain the observed trends in stress, tensile strength, fracture elongation rate, and yield strength as ACL content increases? What are the implications of the enhanced toughness observed in the co-polymers with ACL content below 1wt%? How does the entanglement of chains and rearrangement behavior contribute to the improved tensile mechanical properties?
7. You mentioned that the appropriate addition of ACL can improve the tensile mechanical properties of the co-polymers. Could you provide further insights into the optimal ACL content and its relationship to the observed enhancements? How does the ACL content beyond 1wt% affect the tensile strength and fracture elongation rate?
8. While the results and observations are presented, there is a lack of comparative analysis with previous studies or relevant literature. It would be beneficial to discuss how the findings align or differ from previous research, providing a comprehensive understanding of the contributions made by this study.
Reviewer 3 Report
Comments and Suggestions for Authors
The manuscript is well-written and can be accepted as it is.
Author Response
Thank you very much for taking the time to review this manuscript.
Reviewer 4 Report
Comments and Suggestions for Authors
This is an interesting study with some really good results. While the applied rheological methods are not a solid proof for long-chain branching (extensional rheology would be appropriate), the results look solid enough. The discussion of crystallinity effects is clearly insufficient and requires comparison(s) to other cases of copolymerization and branching effects, like in case of long-chain branched PP or LLDPE.
The following minor issues need to be addressed specifically:
- In the abstract, the word "additionally" seems to be misplaced. If you mention one method as also confirming branching, the first one needs to be mentioned, too.
- Introduction: PA-6 is certainly not a "high-performance polymer" - it belongs to the group of engineering thermoplastics, but it is an established mass product close to commodities.
- For all figures, the color of either the PA-6 or of the 5% copolymer with ACL should be exchanged, as these are too similar and therefore confusing.
- Figure 7: The normal term is "Cole-Cole plot".
- Tables 3 and 3 have excessive post-comma positions, e.g. 219.63°C should be 219.6°C and 312.24% should be 312%. The last parameter of Table 3 is normally called "extension at break (fracture)" or "strain at break".
- Conclusions: A crystallinity change as listed in Table 2 is not an "unchanged" crystalline structure. "Limited changes" might be a better expressio.n
Comments on the Quality of English Language
A good proofing regarding grammar and style would improve readability. Mistakes in punctuation are numerous.
Round 2
Reviewer 2 Report
Comments and Suggestions for Authors
Dear Authors,
I would like to express my appreciation for taking my feedback into account and revising your work. Based on the improvements you have made, I am pleased to recommend the publication of your article in its current form.
Thank you for your hard work and dedication to this project.